# Evaluation of the Performance of Atomic Diffusion Additive Manufacturing Electrodes in Electrical Discharge Machining

**DOI:** 10.3390/ma15175953

**Published:** 2022-08-28

**Authors:** Pablo Bordón, Rubén Paz, Mario D. Monzón

**Affiliations:** Mechanical Engineering Department, Universidad de Las Palmas de Gran Canaria, Edificio de Ingenierías, Campus de Tafira Baja, 35017 Las Palmas, Spain

**Keywords:** additive manufacturing, electrical discharge machining, atomic diffusion additive manufacturing, copper electrodes, material extrusion additive manufacturing

## Abstract

Atomic Diffusion Additive Manufacturing (ADAM) is an innovative Additive Manufacturing process that allows the manufacture of complex parts in metallic material, such as copper among others, which provides new opportunities in Rapid Tooling. This work presents the development of a copper electrode manufactured with ADAM technology for Electrical Discharge Machining (EDM) and its performance compared to a conventional electrolytic copper. Density, electrical conductivity and energy-dispersive X-ray spectroscopy were performed for an initial analysis of both ADAM and electrolytic electrodes. Previously designed EDM experiments and optimizations using genetic algorithms were carried out to establish a comparative framework for both electrodes. Subsequently, the final EDM tests were carried out to evaluate the electrode wear rate, the roughness of the workpiece and the rate of material removal for both electrodes. The EDM results show that ADAM technology enables the manufacturing of functional EDM electrodes with similar material removal rates and rough workpiece finishes to conventional electrodes, but with greater electrode wear, mainly due to internal porosity, voids and other defects observed with field emission scanning electron microscopy.

## 1. Introduction

According to ISO/ASTM 52900:2021 [1] Rapid Tooling (RT) is an application used in additive manufacturing (AM) and intended for the production of tools or tooling components with reduced lead times compared with those of conventional tooling manufacturing. In the context of RT and AM, one relevant application is the development of electrodes for Electrical Discharge Machining (EDM). EDM technology is very common in subtractive manufacturing processes in which, for example, milling machining is not capable of producing several geometric features, such as hollow cavities with sharp edges, ribs, etc. The good quality and final finishing of EDM makes this process very suitable for the mechanical industry, in general, and mold manufacturing, in particular (Figure 1).

EDM is a process in which a metal is removed from the workpiece by applying an electromotive force between the part and an electrode (the tool), both immersed in a dielectric liquid. The application of this force generates a series of discontinuous electrical discharges (called sparks) that produce local temperatures sufficiently high to melt or vaporize the metal in the immediate region of the discharge. During machining, the EDM process by penetration involves nine stages: (1) The nearing of a charged electrode towards the part, producing a discharge between the nearest surfaces (the dielectric liquid avoids the formation of premature discharges, cooling the machined area and removing metal chips); (2) The generation of an “energy column” in the fluid (perforation of the medium) when the electric potential difference is large enough, such that a discharge path is created between the electrode and the workpiece (at this stage the voltage reaches the maximum value, but the current intensity is still zero); (3) The starting of the activation time or impulse time (t_i_), defined as the time where the machining takes place—the higher t_i_, the higher the surface roughness of the part (vaporization begins); (4) The heat increases at the same rate as the current intensity (vaporization continues); (5) During this stage, the instant in which vaporization stops, fusion begins, increasing the contamination of the dielectric; (6) Near the end of the process, the current and voltage are stabilized, the heat and the pressure inside the vapor bubble reach the maximum, and part of the metal starts to be removed from the base part; (7) The system is disabled, both the voltage and current intensity drop to zero (this is the deactivation time or pause time, t_0_, defined as the time between two impulses), the temperature decreases and the melted metal is removed; (8) The collapse of the vapor bubble in the previous stage generates a void which is filled with fresh dielectric, decreasing the contamination and cooling the base part, while the remaining melted metal solidifies forming the so-called “white layer” on the base part; (9) Once the contaminants and damaged dielectric are removed, a small crater is observed on the base part, as well as a slight wearing in the electrode. After this stage a new cycle begins.

To perform an acceptable level of efficiency during these nine stages, the EDM electrode material must not only provide high electrical conductivity, but also a low rate of wear for a good rate of material removal, and good conformability (by CNC machining, casting, etc.). Electrolytic copper and graphite are the most common materials used for EDM electrodes, both being suitable for manufacture by the subtractive procedure. Nevertheless, if the list of potential materials for conventional EDM electrodes is quite limited, the possibilities of the application of AM materials is even more challenging with the existing AM technologies. Several investigations were undertaken to produce EDM electrodes where AM is involved. First, it is important to classify the different approaches depending on how direct or indirect the AM process is. The standard ISO/ASTM 52,900 states that Rapid Tooling can be produced directly by the additive manufacturing process or indirectly by producing patterns that are used in a secondary process to produce the actual tools [1]. The second classification criterion is shown in the standard ISO 17296-2 [2]: (a) Single-step processes, in which parts are fabricated in a single operation where the material properties of the intended product are achieved, and (b) Multi-step processes, in which parts are fabricated in two or more operations where the first typically provides the basic geometric shape within the additive manufacturing machine, and subsequent post-processing enhances the part’s material properties (e.g., using infiltration, sintering or other thermal or chemical modification). Table 1 summarizes several potential procedures to produce EDM electrodes by AM according to both criteria.

Although several of the methods shown in Table 1 were performed to manufacture EDM electrodes, the results were not always promising. Several approaches based on direct or single-step processes using PBF were carried out. Meena and Nagahanumaiah [3] investigated the feasibility of using Direct Metal Laser Sintering (DMLS) to produce EN24 steel electrodes. They optimized the EDM process parameters but found that the excessive wear of the electrode and the high level of porosity limit the application. Sahu and Mahapatra [4,5] tested the behavior of AlSi10Mg electrodes made by PBF and tested on the EDM of titanium. They concluded that, in terms of surface characteristics of the part (roughness, surface crack density, white layer thickness, microhardness), the results were acceptable, but in terms of the material removal rate (MRR) and tool wear rate (TWR), the results were worse than using conventional electrodes, such as graphite. Other authors, such as Uhlmann et al. [6], used PBF to produce tungsten carbide electrodes and the EDM results showed that the MRR and the TWR were worse in comparison with conventional copper and graphite electrodes (mainly due to the porous structure created by the AM process). Due to the relevance of a high electrical conductive material as part of the EDM electrode, several studies were carried out with copper as part of the composition. For example, Czelusniak et al. [7,8] used PBF to sinter a composite formed by a metallic matrix (Cu–Ni) and a ceramic (ZrB2, TiB2). Both ceramics are known to be ultrahigh-temperature materials. The overall performance of the composite electrodes was inferior to the conventional Cu electrodes. Uhlmann et al. [9] produced electrodes by PBF of tungsten carbide–cobalt WC-C 83/17, with positive results regarding MRR, but significantly higher TWR than for the solid tool electrodes. However, they observed that the inclusion of internal channels for flushing reduced the relative tool wear from 5.9% to 3.9%.

Copper powder is not easy to process by PBF due to its high reflectivity and high thermal conductivity. However, Uchida et al. [10] succeeded in obtaining almost 99.7% relative density for Cu-1.3 mass% Cr (Cu–Cr) alloy powder. Based on this, Yanagida et al. [11] developed an EDM electrode by PBF of Cu-1.26 mass% Cr (Cu–Cr) alloy powder. To improve the thermal conductivity, the AM part was annealed at 600 °C (increasing the thermal conductivity of the pure copper electrode from 21% to 84%). They also included internal channels for the EDM fluid, increasing the efficiency of the process. The EDM test showed similar behavior in the AM annealed electrode to that of the conventional copper electrode.

Several approaches to indirect AM or multi-step AM, in which a previous pattern is required, were investigated. The pattern can be produced by AM of plastics, such as material extrusion (MEX), powder-bed fusion of plastics (PBF-P), vat photopolymerization (VAP), etc. In the second stage, the pattern must be coated with another material of sufficiently high electrical conductivity and thickness to be suitable for EDM [12]. Alamro et al. [13] performed a two-stage process starting with an ABS AM pattern by MEX (generally known as the FDM process) coated with a 1 mm layer of copper after an electroplating process. This plating was performed after making the ABS electrically conductive using conductive silver paint and introducing the pattern into an electrolytic bath of CuSO_4_ + H_2_SO_4_, under a current density. The EDM test showed good behaviors in terms of TWR and MRR, when compared with a conventional copper electrode. In this process, problems such as pitting or nodular growth (Figure 2) must be reduced using additives in the bath and controlling the process parameters during the deposition, improving the surface quality. A similar approach combining electroless plating (no current density) and electroplating was developed by Padhi et al. [14]. In this research, the maximum thickness of copper was 0.2 mm. To avoid the problem of pitting and nodular growth, and to create a thicker copper wall and better surface quality, the procedure of electroforming was shown to be more effective. Several studies applying electroforming to make EDM electrodes were undertaken. Electroforming uses a typical electroplating process but removes the metal shell from the pattern once the electroplating ends, thereby producing a high-quality surface reproducing the roughness and small features of the base model. The high level of reproducibility of an electroformed electrode is even suitable for micro-EDM, such as shown by Sánchez et al. [15]. They produced an electroformed EDM copper electrode starting with a resin pattern made by Digital Light Processing AM (DLP, Vat Photopolymerization), resulting in an electrode more suitable for EDM finishing conditions, rather than roughing, due to the increase in the TWR. Another procedure for making EDM electrodes using an indirect AM process is cold spraying of copper [16]. This process allows the production of near-net-shape metal parts via supersonic 3D deposition of powder onto a 3D model. A different indirect AM approach was carried out by Wu et al. [17] by investment casting technology. They produced a wax pattern by AM to make a sand mold for casting with melted brass. The resulting electrode was suitable for the EDM rough-machining process and for quicker manufacturing, in which several electrodes are produced simultaneously in an assembly tree mold.

As shown in the above literature review, several studies analyzed the behavior of indirect AM electrodes for EDM, but there are no references regarding the performance of multi-step AM electrodes. Among the list of potential procedures to produce EDM electrodes using multi-step AM (Table 1), the “material extrusion of composite–debinding–sintering process” should be highlighted, as there are two relatively new commercial technologies available in the market based on this concept. Technically speaking, one of these multi-step AM processes is known as Atomic Diffusion Additive Manufacturing (ADAM), and it represents a promising method for the production of metal parts, especially due to its reduced costs compared with other direct AM technologies for metal parts. Despite this potential, the performance of electrodes made with ADAM technology, and intended for EDM applications, was not investigated thus far.

In this work, this alternative AM multi-step process (Atomic Diffusion Additive Manufacturing, ADAM) was used to produce copper electrodes and their performance in EDM evaluated, thus filling the gap found in the literature. With a reference electrolytic copper electrode, experiments were designed and performed with different EDM process parameters (average intensity, impulse time, pause time and capacitor level), resulting in 16 machining strategies. For each strategy, the resulting roughness, stock removal rate and electrode wear rate were assessed. The results were used in an optimization algorithm to determine the best parameter combination to minimize the electrode wear rate or maximize the stock removal rate for different roughness values. The selected EDM strategies were applied in the ADAM copper electrode and in the standard electrode, and the results were compared.

## 2. Materials and Methods

### 2.1. Materials

Two EDM electrodes were manufactured with two types of copper raw material: electrolytic copper (electrolytic electrode) and filament copper composite (ADAM electrode).

Electrolytic copper was provided as a square solid bar with a purity of 99.9%, for the manufacturing of the conventional electrodes by machining processes. This material is specifically produced for EDM electrodes for its purity and electrical properties.

On the other hand, Markforged copper filament F-MF-1010 was used for the manufacture of electrodes using ADAM technology. The copper filament is a mixture of powder copper and a binder, which is removed in the last stages of the part-forming process (washing and sintering stage). The copper pieces finally obtained must contain a copper composition greater than 99.8%, according to Markforged, with a maximum concentration of 0.05% oxygen and 0.05% iron. The data sheet for the copper filament shows that, once the final part is manufactured, it should have an ultimate tensile strength of 193 MPa, 98% relative density and 84% electrical conductivity with respect to the Annealed Copper Standard (100% IACS). As support material in the printing stage, Markforged ceramic release F-MF-1002 Type-1 filament was used. 

For the washing step of the electrodes manufactured with ADAM technology, Opteon SF79 Specialty Fluid solvent was used. 

Finally, 12 workpieces of 50 mm diameter and 16 mm height were machined from aluminum 2030 (samples to erode).

### 2.2. Electrode Manufacturing Method 

The electrodes were designed as a parallelepiped geometry, 31 mm in height, 28 mm in width and 9 mm in length, in which the surface used for erosion corresponded to a 28 mm × 9 mm face. 

The electrolytic electrode was mainly manufactured by a universal CNC machine XYZ SMX3500. The face to be used for the EDM was finally rectified in a GER S-40/20 machine with a ceramic–corundum grinder, Tyrolit HA54VG. 

The second electrode was manufactured by AM in the Markforged Metal X^TM^ System. This system is an innovative AM technology with some similarities to material extrusion AM (MEX), with the initial part (green part) being obtained by the selective deposition of a filament material (metal powder in a plastic matrix). This technology is a less expensive alternative than other metal AM technologies. Moreover, it is more economical than casting or machining metals, particularly in short productions, especially for complex parts that are unfeasible for conventional production methods [18]. The Metal X^TM^ System is made up of 3 stages: 3D printing, chemical debinding (wash) and sintering.

After completing the 3D design using CAD software, the first manufacturing stage was carried out in the Metal X Printer (second generation). The electrode was printed in a vertical orientation, using the process parameters shown in Table 2. The printed electrode (green part) was not yet a final part since it was composed of a mixture of copper powder (not yet sintered) and wax and polymer binding components. To compensate the shrinkage of the part during the process, the software automatically scaled the part in a range from 14% to 16% (for this copper part).

Despite the triangular infill, note that the surface of the electrode used to erode was completely dense (upper layers of the 3D printing, 1.01 mm thickness). Most of the printing parameters were not configurable in this system, so the part orientation, type of fill, and wall thicknesses were established manually. The remaining parameters were preset by the technology. The electrode was manufactured in 4 h and 23 min and weighed 57.33 g.

The second stage corresponded to the chemical debinding process. The washer is a solvent-based debinding system in which the green part is immersed in a specific solvent which removes the primary binding material (wax). The process was carried out in a Markforged Wash-1 machine for 12 h. After washing, the part was also dried in the drying chamber of the Wash-1 machine for 4 h (no additional time was needed), where it was air-dried to remove the solvent before sintering. After debinding and drying, the weight of the electrode was 55.68 g. The brown part, therefore, lost 2.88% mass compared with the green part, which is in compliance with the minimum mass-loss threshold value for copper material during the washing step (2.7%). Otherwise, the washing step must be repeated until the mass loss of the brown part is at least 2.7% compared with that of the green part.

Despite the washing process, a small amount of the remaining wax binder, as well as the polymer binder, were removed during the final sintering stage (the part is semi-porous after washing so the remaining binder can be easily burnt off during sintering). This final process was completed in the Markforged Sinter-1 furnace. In this phase, the sintering process converted the brittle brown part in an entire metal part by heating it to approximately 85% of the melting temperature of the metal, in a mixture of inert gases (argon and nitrogen). During the initial heating phase, the furnace burned and extracted the remaining binder through the porosity created by the washing process. At the peak of heating, the electrode shrank to its final size, while the ceramic supports turned to dust. Finally, a slow cooling was carried out to extract the finished part (note that the thermal treatment was also preset by the technology for each material). The total sintering process was 27 h and 13 min. Finally, the raft was removed and the ADAM electrode cleaned. Figure 3 shows the manufacturing complete process of a real batch of copper parts, including the ADAM electrode.

However, for a reliable comparison, the ADAM electrode surface to be used in EDM was also rectified through the same process and machine as the electrolytic electrode.

### 2.3. Density Measurement

ADAM technology theoretically allows the manufacturing of 100% dense metal parts. However, the printing and sintering stages produce parts of lower density than electrolytic raw material. To accurately compare the EDM process in both electrodes, density measurements were carried out in a Metrotec MDS-300 densimeter. In the case of the electrolytic copper electrode, the measurement was made directly on the electrode itself. For the density measurement of the ADAM electrode, which was made with non-complete dense infill (as an additive manufacturing advantage), three cubic copper samples were manufactured with ADAM technology (Figure 4), with 100% infill density and 10 mm sides.

### 2.4. Conductivity Measurement

The volumetric and surface electrical conductivity of both copper materials was performed by a Keithley 2400 Source Meter (Mitutoyo Corporation, Kawasaki, Japan). For these measurements, the cubic parts manufactured by ADAM were used. For the electrolytic material a cubic piece with similar dimensions was manufactured.

The volume electrical conductivity (*σ_V_*) and the surface electrical conductivity, per square, ($\sigma_{s}$), were calculated from the electrical resistance following Equation (1).
(1)$$\sigma_{V}=\frac{1}{R}\cdot \frac{l}{A_{c}}\;\;\;\;\;\;\;\;\;\;\;\;\;\sigma_{s}=\frac{1}{R}\cdot \frac{l}{P_{c}},$$
where ***R*** is the measured electrical resistance, ***l*** is the length between electrical contacts, $A_{c}$ is the electrical contact area for volumetric conductivity and $P_{c}$ is the electrical contact perimeter.

### 2.5. Energy-Dispersive X-ray (EDX) Spectroscopy Analysis and Scanning Electron Microscope (SEM)

The surface chemical composition of the two types of electrodes manufactured was analyzed using a Hitachi TM 3030 scanning electron microscope (SEM) at an acceleration voltage of 15 kV (Hitachi Ltd., Tokyo, Japan) coupled with an EDX detector. 

The results obtained from the EDX analysis included the identification of the elements present in a specific area of the part using the mapping tool, as well as the quantification of those elements. Five measurements were taken on each sample for quantification purposes, analyzing different regions of the parts.

A Zeiss Sigma 300 VP field emission scanning electron microscope (FESEM) (Carl Zeiss AG, Jena, Germany) was also used for high-quality image analysis. 

### 2.6. Roughness Measurement

Arithmetical mean roughness (Ra) measurements were carried out on the 3 eroded areas of each strategy by using a Mitutoyo SJ-201P surface-roughness tester. Data acquisition software was used to set and perform the measurements. After each EDM process, 3 measurements were taken along the length of the eroded hole, in the lower, middle and upper parts (Figure 5). The measurement process was configured according to ISO 1997, with a 2.5 mm cutoff length, 3 sampling lengths, and profile R with 2CR75 filter. The obtained values were then used to determine the VDI parameter according to the equation VDI grade = 20 log (10 Ra) [19], where Ra is the arithmetic average roughness in µm. The VDI parameter is widely extended as a standard for roughness characterization in EDM processes. The average value of the three VDI measurements was calculated.

### 2.7. Electrical Discharge Machining Tests and Optimization

EDM processes were conducted in an ONA DB300 EDM machine. In this EDM machine, there were no EDM parameters available to erode aluminum parts with a copper electrode; to address this, a previous 2-level full factorial design of experiments was performed with the electrolytic electrode on the aluminum workpiece, and the data obtained were fed to an optimization algorithm that allowed the optimal parameters to be found. To do this, the most influential parameters in the EDM process with a copper electrode and a steel workpiece were first analyzed. 

A key parameter of the EDM process is the resulting surface roughness of the eroded workpiece, normally measured as VDI grade. VDI grades from 16 to 22 (copper electrode on steel workpiece) were established as a reference for its typical applications in mold manufacturing, for which the EDM process is very useful. These VDI values were used simply as a reference to select the variable parameters, their limit values and the fixed values of the remaining parameters. Note that despite this preselection of VDI, the resulting VDI value may change as the workpiece material is different (aluminum instead of steel) and this also applies to the combination of process parameters. According to this preliminary analysis, the following parameters were set as fixed values as recommended by the EDM machine manufacturer for the use of a copper electrode on a steel part (Cu-St) for the VDI grade selected:1.Ionization Voltage: −200 V;2.Erosion servo: 65 V;3.Return time: 0.3 s;4.Work time: 0.3 s;

On the other hand, the erosion depth was set to 0.18 mm, the flow level was kept at level 15 and the slurry pumps at 60 s. In view of the manufacturer’s recommendations to reach a VDI grade between 16 and 22 with a copper electrode working on an aluminum part, the following variables and limits were established:5.Average Intensity: 0.5 A (level 1)–2 A (level 3), with level 2 corresponding to 1 A;6.Impulse time: 3.2 μs–6.4 μs;7.Pause time: 3.2 μs–6.4 μs;8.Capacitor: 2–13.

Once the process parameters and their maximum and minimum limits were defined, the 2-level full factorial (2^n^) design of experiments was performed with 3 replicas for each of the 16 EDM strategies. The average results of the response variables (explained in the following sections) for each strategy were also calculated. This allowed the checking of the repeatability of the test to identify possible outlier values. Table 3 summarizes the variables and fixed values of the experiment design.

The resulting strategies of the 2-level full factorial design of the experiment are shown in Table 4.

Once the table with the post-EDM experimental results was completed (depicted in Section 3.2), the available data was input to an optimization algorithm based on genetic algorithms to find the best combination of process parameters to minimize the electrode wear rate and maximize the stock removal rate. 

The optimization algorithm was based on a genetic algorithm with 100 generations, each one with 100 individuals. The first population is randomly created. The algorithm then evaluates the fitness function of each design. In this case, this evaluation was carried out by using an interpolation method (Kriging metamodel) [20] for each response variable. The metamodels were created from the data gathered in the previous experiment designs. An exponential correlation model and a polynomial regression model are used in the Kriging metamodel, with the peculiarity that the polynomial regression model varies from order 2 up to 0 depending on the available data. The 2-order regression model can obtain better estimations but requires more sampling or a better distribution of them. The lower the order, the easier the creation of the metamodel, but the lower the accuracy. For this reason, the optimization algorithm includes a loop that initially uses the 2-order regression model for the prediction of each response variable and, if the generation of the metamodel fails, the 1-order regression model is applied automatically (and so on for the 0-order). Therefore, the best regression model according to the available data is always used [21].

The fitness function is calculated using the corresponding metamodel for each response variable. In this case, the fitness function is the corresponding objective (electrode wear rate or stock removal rate) with a penalty factor that penalizes the fitness value if the optimization constraints (VDI values according to the metamodel predictions) are not fulfilled. Once the fitness function is calculated for each individual, the genetic algorithm applies a tournament selection of 2 individuals, an arithmetic crossover, mutation, reparation and elitism.

Since the selected design variables are limited to discrete values in the EDM equipment, the genetic algorithm was prepared accordingly to work with discrete variables. Table 5 shows the configurable values in the EDM equipment, and the limit and possible values established for the optimization algorithm.

For the crossover in the genetic algorithm, and for each design variable, a random value is generated (between 0 and 1) and, if the value is lower than or equal to 0.5 (50% crossover probability), the first offspring will take the value of the first parent and the second offspring will take the value of the second parent. Otherwise, the first offspring will take the value of the second parent, and the second offspring will take the value of the first parent.

For the mutation, for each individual of the resulting population after the crossover, a random value is generated (between 0 and 1). If this value is lower than or equal to 0.6 (60% mutation probability), the corresponding individual is mutated. The mutation consists in selecting a random design variable and slightly modifying the current value by adding the result of a random value between −0.5 and 0.5 multiplied by the maximum interval of that design variable. In the case of the discrete variables, a round operation is applied to obtain an integer value.

This algorithm was applied to a preliminary optimization process to minimize the electrode wear rate, and to maximize the stock removal rate, always keeping the VDI value below 24. Note that despite the reference VDI values being between 16 and 22, the resulting VDI values after the experiment designs were between 22 and 26, as depicted in Section 3.2. This difference may be normal as the materials used were different (ADAM copper on aluminum workpiece compared with standard copper on steel workpiece); this is also the case for the combination of process parameters. In the maximization of the stock removal rate, the algorithm led to an optimum of level 3 for average intensity, 6.4 μs impulse time, 3.2 μs pulse time and level 6 for the capacitor. This new strategy was also carried out experimentally and the results were added to the design of experiments as strategy 17 (depicted in Section 3.2).

With these 17 data, the minimization of the electrode wear rate and the maximization of the stock removal rate were applied with 3 VDI ranges established as constraints for the optimization process: VDI 23 (22.5 < VDI < 23.5), VDI 24 (23.5 < VDI < 24.5) and VDI 25 (24.5 < VDI < 25.5).

### 2.8. Stock Removal Rate

The stock removal rate was calculated (as shown in Equation (2)) by relating the material removed from the part, weighed before and after EDM, to the duration of the process. To obtain a volumetric result, the differential weight was converted to mm^3^ using the measured aluminum density.
(2)$$Stock\;removal\;rate\;\left(mm^{3}/min\right)=\frac{Initial\;part\;weight\;\left(g\right)-Final\;part\;weight\;\left(g\right)}{aluminum\;density\;\left(g/mm^{3}\right)\cdot \;Work\;EDM\;time\;\left(min\right)},$$

### 2.9. Electrode Wear Rate

To determine the wear rate of the electrodes, both the volume removed of the part and the volume of electrode worn were calculated, as shown in Equation (3). This parameter allows the wear of the electrode related to the volume of erosion of the stock to be obtained, regardless of the required time in the EDM process.
(3)Electrode wear rate %=Initial electrode weight g−Final electrode weight gcopper density g/mm3Initial workpiece weight g−Final workpiece weight galuminum density g/mm3·100,

Figure 6 summarizes the methodology applied.

## 3. Results and Discussion

### 3.1. Density and Electrical Conductivity Results

The densities of the electrodes and workpieces are shown in Table 6. Regarding the ADAM electrode, the measured density was around 95.1% of the standard copper density (8.5198 g/cm^3^ for ADAM copper, and 8.9587 g/cm^3^ for electrolytic copper), slightly lower than the Markforged specifications (98%). This difference is similar to that found by Galati et al. in other density analyses of parts manufactured with ADAM technology [22], which is related to the internal porosity of the ADAM copper (more details about this porosity will be addressed at the end of Section 3.5).

Table 7 shows the electrical conductivity results for both samples. Similar volumetric and surface electrical conductivities were obtained (note that despite the difference in the absolute values, the results are in the same order of magnitude, meaning similar results in terms of electrical conductivity).

### 3.2. SEM–EDX Results

The result obtained by mapping the surface of the ADAM samples is shown in Figure 7. The images correspond to the identification of elements that are present in the part. According to the results, and apart from cooper (Cu), carbon (C) and oxygen (O) were also detected on the analyzed surface. These three elements were identified in the measurements carried out with both samples tested.

On the other hand, Figure 8 shows the quantification results obtained in one the measurements carried out, while the results from the quantification analysis of both types of parts are summarized in Table 8.

As shown in Figure 8, different peaks attributed to the identification of Cu on the surface of the tested parts can be seen. This result is explained by emission of different X-rays that are characteristic of this element and can be an indicator of the existence of different types of copper-based species, observed in all the tested parts. In this way, the detection of oxygen on the surface of the parts (especially on the gaps and cavities observed in Figure 9) could be related to the presence of copper oxides. Very similar EDX spectra were obtained by Kooti et al. [23] when analyzing Cu_2_O nanoparticles, which supports the hypothesis of copper oxides being present on the part’s surface. On the other hand, the carbon percentage observed is attributed to impurities located in the cavities and voids of the parts, mainly due to their handling during manufacturing and characterization.

The statistical analysis of the obtained data was performed using MATLAB software (MATLAB and Statistics Toolbox Release 2021a, The MathWorks, Inc., Natick, MA, USA) and by comparing the two groups using the Wilcoxon two-sided rank sum test. Highly statistically significant differences (*p* < 0.01) were obtained when comparing the amount of oxygen quantified in both groups. Apart from the content of oxygen being significantly higher in the case of the part manufactured using the ADAM technology, no significant differences were observed regarding the rest of the elements identified on the surface of the samples analyzed by EDX.

### 3.3. Results of EDM Test Experiments

Table 9 summarizes the parameters of the 17 EDM strategies carried out with the electrolytic copper electrode and aluminum workpiece (2^4^ and one additional strategy from the preliminary optimization), and the average results (VDI, electrode wear rate and stock removal rate) of the three replicas of each strategy. More information about the results of the EDM and roughness tests can be found in Appendix A.

As expected, the variation of the three established output parameters is quite heterogeneous, increasing or decreasing unevenly, being affected depending on the modification of the variable parameters, since in the EDM process each process variable produces different effects in the results. Because the variables of the process affect each of the results differently, none of the strategies simultaneously optimized these three output parameters or response variables. Moreover, although the optimization algorithm is responsible for handling this information to obtain the best configurations, a brief analysis of the results obtained was carried out.

As shown in Table 9, the average electrode wear rate for the lower intensity level was slightly higher than that for higher intensity levels, while the material removal rate was the opposite. At both intensity levels, the VDI grade was similar (around 24).

The most notable differences in electrode wear rate were observed when the capacitor level was modified, being more pronounced in the cases of a lower intensity level (strategies 1–8). In relation to the stock removal rate, no specific trends were observed.

Regarding the VDI grades, lower capacitor levels (odd strategies) led to better VDI values in practically all cases compared with capacitor levels of 13 (even strategies). This can be easily seen by comparing each pair of strategies for which only the capacitor level is modified. 

Regarding the impulse and pause times, higher electrode wear rates, and also higher stock removal rates, were observed when working with a pause time of 3.2 μs and the capacitor at level 2. However, for a capacitor at level 13, the 6.4 μs pause time produced higher stock removal rates, while maintaining similar electrode wear rates. No significant effect of the impulse time was observed when working with the capacitor at level 2. In the case of the capacitor at level 13, the wear rates were very similar, regardless of the remaining parameters, and a slightly better stock removal rate was also observed for a shorter impulse time.

Strategy 3 produced the minimum electrode wear rate (4.979%) and one of the lowest VDI values (23.60). The minimum VDI grade was obtained by strategy 5 (22.62) with an electrode wear rate of 7.29%. Regarding the stock removal rate, strategy 10 allows the highest value (0.813 mm^3^/min) to be obtained but with a high electrode wear rate (9.753%) and a VDI of 25.41.

The established strategies do not cover all the possible combinations of the selected process variables. Therefore, there may be untried strategies that could obtain better results. In this regard, the optimization algorithm leads to the estimated optimal combinations from the data gathered, which may be, in some cases, different to the strategies already evaluated.

### 3.4. Results of Optimal Strategies Calculated

For the preset VDI and the chosen optimization objectives, the optimal variables requiring configuration in the EDM process were estimated by the optimization algorithm developed. Table 10 shows the summary of the optimal design variables and the estimated VDI, electrode wear rate and stock removal rate for these combinations of design variables.

The results of the optimization processes show that, for VDI 23, the optimal configuration for the minimum electrode wear rate coincides with strategy 15, and the optimal for the maximum stock removal rate coincides with strategy 9. For VDI 24, the optimal for the minimum electrode wear rate matches strategy 3. For VDI 25 and maximum stock removal rate, the algorithm selected strategy 10 as the optimal combination. Nevertheless, in the remaining cases, the algorithm converged to new strategies (for VDI 24 and maximum stock removal rate, new strategy 18; and for VDI 25 and minimum electrode wear rate, new strategy 19).

### 3.5. Results of New EDM Machining with Optimal Strategies

The two new strategies (18 and 19) obtained by the optimization algorithm were experimentally tested. Table 11 shows both the estimated results (by the algorithm) and the real results. 

Regarding strategy 18, where a maximum stock removal rate for VDI 24 was expected, the results showed a great improvement in the stock removal rate, higher than the double expected, and a similar electrode wear rate to the minimum wear rate strategy (4.94%); however, a higher VDI was obtained. Despite this difference, strategy 18 was considered of interest for the analysis of the performance of both electrodes since it had the highest stock removal rate obtained so far. 

For strategy 19, the minimum electrode wear rate was widely achieved (more than half of the expected value) and improved the stock removal rate about six times more than expected. However, as in the previous strategy, the VDI also increased. For this reason, although the actual VDI ranges of both strategies do not correspond with the initially set VDI range, both strategies were also selected for the final comparison of the electrolytic and ADAM electrodes.

It should be noted that the estimations of the optimization algorithm were limited in terms of accuracy due to the low number of data points (17). In the case of the capacitor, the search space ranged from two to thirteen (discrete values of 1 step) and the design of experiments only tested the extreme values of this design variable. Other types of experiment design, such as the Latin Hypercube, could have been used, but since the Kriging interpolation method is more accurate in interpolated estimations than in extrapolations, the full 2^n^ design of experiments was selected. Note that this optimization algorithm could be applied to additional iterations, adding the new data obtained (strategies 18 and 19) to the database, which would improve the accuracy of the estimations compared with the real results. In this case, the objective of the optimization algorithm was to select a number of interesting strategies for the comparison of the two types of electrodes.

The final six strategies selected for the EDM machining comparison with both electrodes correspond to strategy numbers 3, 9, 10, 15, 18 and 19. These strategies include a wide range of parameters (and results, according to the tests with the electrolytic copper electrode) that allows for a relatively complete comparison between both electrodes. 

### 3.6. Results of EDM Machining with ADAM Copper Electrode

The EDM machining with the ADAM electrode (Figure 10) was performed for each selected strategy and the results compared to those obtained by the electrolytic electrode. Table 12 shows the results obtained (sorted from the lowest to the highest VDI obtained, for the sake of clarity).

Figure 11 shows the VDI values for both the electrolytic and ADAM electrodes. The results were quite similar, with a slightly higher roughness for the ADAM electrode in all the tests except test number 1, where the roughness was minimally lower by around 2.71%. In the remaining cases, the VDI of the electrolytic electrode with respect to the ADAM electrode was between 1.19% and 3.35% lower. The higher roughness obtained with the ADAM electrodes may be due to the presence of voids and non-sintered particles in the internal structure (more detail about this will be found at the end of this section). As the electrode wears out, these defects are exposed in the electrode surface and, consequently, the defects are engraved in the workpiece, thus leading to higher roughness. Additionally, the non-sintered particles are released more easily and may reduce the quality and stability of the sparks, also increasing the resulting roughness. In any case, these minimal differences indicate a very similar behavior between electrodes in relation to the roughness obtained, despite the use of AM technologies, which produces solid layers of material but with structural and internal defects between layers [24].

In terms of stock removal rate (Figure 12), the behavior of both electrodes was very similar, with trends very close to each other and without a clear predominance of the results of one electrode over the other. As shown in Figure 12, the difference in the last test was minimal (less than 0.5%). However, in tests 2, 4 and 5 the stock removal rate of the ADAM electrode increased between 4.9 and 9.2% compared with the electrolytic electrode, while in test 1 the electrolytic electrode increased the stock removal rate by 20.5% compared with the ADAM electrode. These results indicate that there is no general pattern of improvement or worsening according to the chosen electrode, largely dependent on the parameters of the EDM processes, which can influence the behavior of the electrodes in a disparate way.

Finally, Figure 13 shows the behavior of the electrodes in terms of wear rate. In this case, the wear rate of the ADAM electrode was more pronounced than that of the electrolytic electrode in practically all the tests except for number 5 (almost identical wear rates). In the remaining tests, the wear rates of the electrolytic electrode were close to half of the wear rates of the ADAM electrode, with values that ranged between 40.8% and 52.1% of those of the ADAM electrode. 

According to the results, therefore, the wear rate of the ADAM electrode is higher than that corresponding to the electrolytic electrode. This behavior may be due to a lower internal cohesion of the material, probably caused by the internal structures and the defects generated both during the printing process and in the final sintering stage, generating structures that are not as homogeneous or consistent as those of the electrolytic materials manufactured by casting. Other authors [24,25] already reported defects in parts generated with ADAM technology, highlighting the lack of union between layers and adjacent filaments deposited during the printing process (and not completely resolved during the sintering stage), as well as the internal porosity and orientation effects of the layer structures. As the electrode wears out during the EDM process, these regions with lower adhesion are exposed and released more abruptly (clusters), thus increasing the electrode wear rate compared to the electrolytic copper electrode, which is a more homogeneous material. Additionally, the release of non-sintered particles and clusters may reduce the quality and stability of the sparks, thus increasing the resulting roughness. 

Figure 14a shows a lateral surface of the ADAM electrode, where the layered structure is clearly observed, as well as the granulate structure of the sintered metal. In Figure 14b, the copper spherical material with heterogeneous particle sizes and voids between particles is clearly identifiable. However, this surface appearance was expected since the external faces do not have adjacent material for a complete sintering.

For the analysis of the internal regions, one of the cubic copper samples manufactured by ADAM was fractured. The internal structure is shown in Figure 15. In the image on the left, two different regions can be observed: the majority region with fracture marks and the inter-filament lines (marked with arrows, corresponding to areas with a lack of material between adjacent filaments). The image on the right shows the details of the framed area of the picture on the left, where the differences between both zones are more evident. The fractured areas show a large number of medium-sized voids, probably increased by the fracture process, but also micro-voids and non-sintered particles, evidencing the internal porosity and the non-completely dense structure of the material. Furthermore, in the areas between adjacent filaments there are no fractured edges (unlike the other region), but larger voids without material. This defect may be due to a lack of deposited material during the printing process which is not completely removed during sintering.

Additionally, the sintering process was not fully completed (Figure 16). A large number of micro-voids were created, mainly in the bonding interphases between particles (indicated by the arrows), perhaps produced by the process of polymer debinding from the innermost areas during the sintering process.

## 4. Conclusions

This work presents the results of the development of copper EDM electrodes manufactured using Atomic Diffusion Additive Manufacturing technology and their EDM behavior compared to conventional electrodes made of electrolytic copper. Despite the potential of the ADAM technology for the manufacturing of metal parts, this is the first study that addresses the application and evaluation of the performance of copper ADAM electrodes in EDM.

The results of density and electrical conductivity of the electrode manufactured with the ADAM technology showed similar values to electrolytic copper, although with a density slightly lower than that specified by the developer of the technology.

EDM test experiments with an electrolytic copper electrode on aluminum parts, combined with a design of experiments and optimization algorithm, allowed six selected EDM strategies to be obtained. These were applied for both the electrolytic and ADAM electrodes and the subsequent comparison of results between them. The optimization algorithm showed a wide potential to continue obtaining new EDM strategies based on the desired optimization objectives (minimum electrode wear rate or maximum stock removal rate).

The results of the selected EDM tests showed that ADAM technology enables fully functional copper electrodes to be obtained, with stock removal rates and roughness (VDI) of the final workpiece very similar to those of electrolytic copper. The ADAM electrode led to a slightly higher VDI than electrolytic copper due to the exposure of voids and defects during the EDM process, which were then replicated in the workpiece, thus increasing the roughness. Additionally, as the electrode wore out, the release of non-sintered particles and clusters may have reduced the quality and stability of the sparks, thus increasing the resulting roughness. In any case, the increase in VDI was very low compared with that obtained with the electrolytic copper electrode. This opens up a promising new line of development of non-conventional copper electrodes through AM, which makes it possible to develop highly complex geometries that are difficult or unfeasible to obtain by conventional technologies.

In contrast to the favorable results of the stock removal rate and VDI, the wear rate of the electrode manufactured with the ADAM technology was much higher than that of the conventional electrode, mainly caused by defects and internal structures generated by the AM process. As the electrode wore out during the EDM process, these regions with lower adhesion were exposed and released more abruptly (clusters), thus increasing the electrode wear rate compared with that of the electrolytic copper electrode. These wear rates represent one of the main challenges requiring a solution in terms of the development of new electrodes manufactured by ADAM. Improvements could be achieved, for example, by optimizing the part-printing strategies or sintering process temperatures [26], which could significantly improve several defects that could cause the higher electrode wear rate.

Finally, taking into account the higher wear rate of the electrodes developed by the ADAM technology, these electrodes could have a great interest as roughing electrodes with high rates of stock removal, where the electrode wear rate is not a conditioning factor of the processes. Additionally, similarly to conventional EDM processes, the use of a second finishing electrode would minimize this negative aspect, thus being feasible to develop any EDM process with electrodes manufactured by ADAM technology. In any case, it should be noted that this study focused on copper ADAM electrodes (Metal X^TM^ System) working on an aluminum workpiece. Therefore, these conclusions may vary for other electrodes or workpiece materials, and even for other ADAM technology.

## Figures and Tables

**Figure 1 materials-15-05953-f001:**
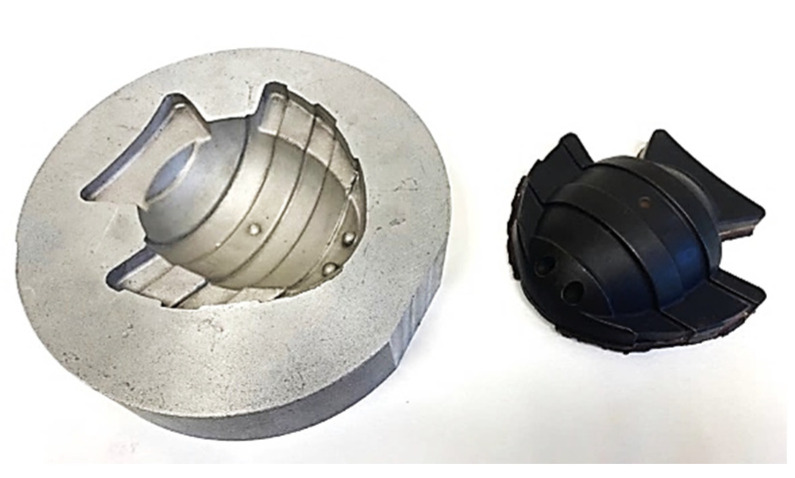
Mold manufactured by EDM (**left**) and EDM electrode (**right**).

**Figure 2 materials-15-05953-f002:**
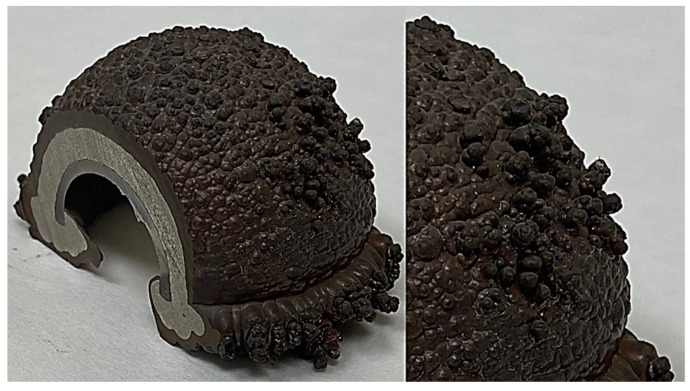
Nodular growth defects.

**Figure 3 materials-15-05953-f003:**
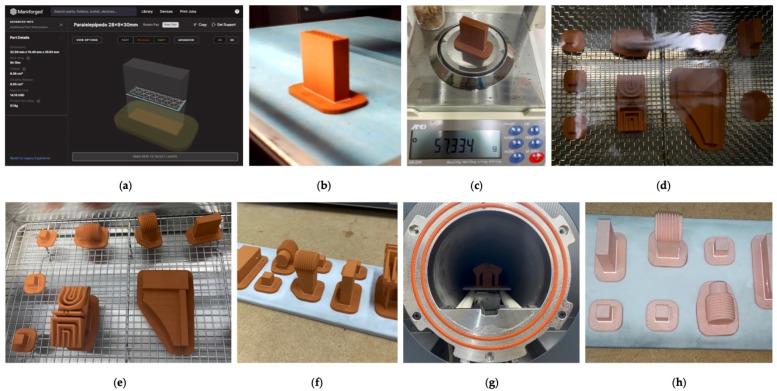
Manufacturing scheme for a batch of parts using ADAM technology: (**a**) Electrode design; (**b**) Printing; (**c**) Initial weight; (**d**) Debinding; (**e**) Drying; (**f**) Pre-sintering; (**g**) Sintering; (**h**) Final parts.

**Figure 4 materials-15-05953-f004:**
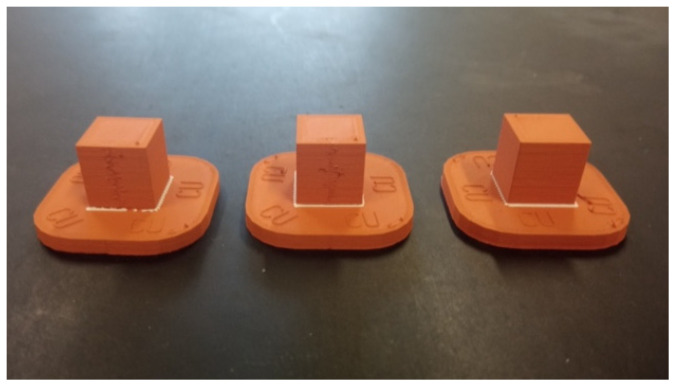
Cubic test samples manufactured by ADAM technology.

**Figure 5 materials-15-05953-f005:**
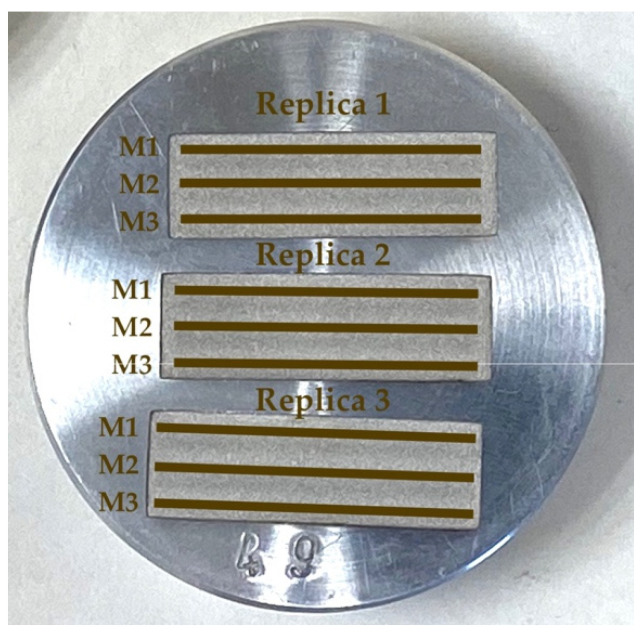
Roughness measurement lines in workpiece.

**Figure 6 materials-15-05953-f006:**
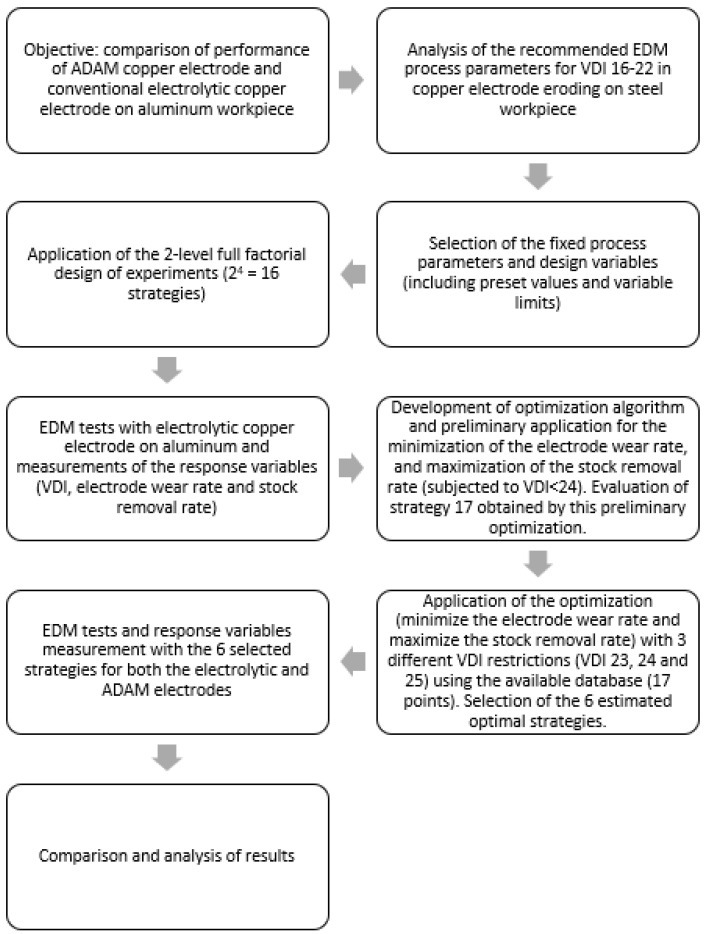
Summary workflow of the methodology applied.

**Figure 7 materials-15-05953-f007:**
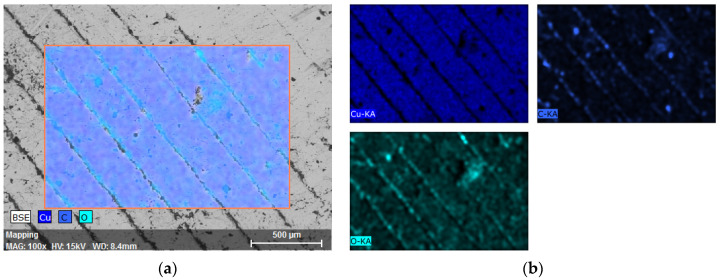
SEM–EDX analysis of the ADAM sample: (**a**) Region selected; (**b**) Identification of elements present on the surface.

**Figure 8 materials-15-05953-f008:**
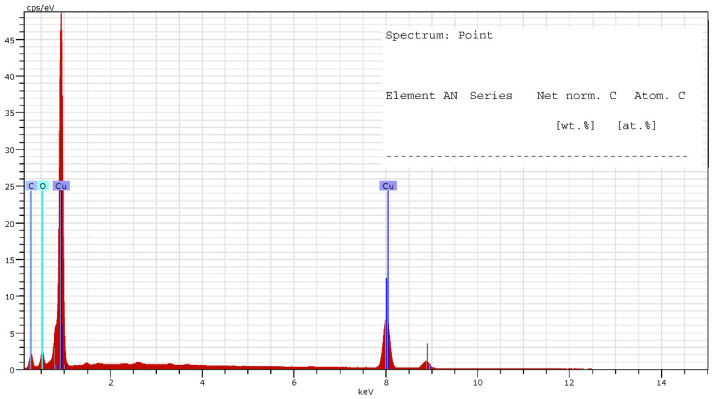
Spectrum of elements on the surface of the ADAM electrode.

**Figure 9 materials-15-05953-f009:**
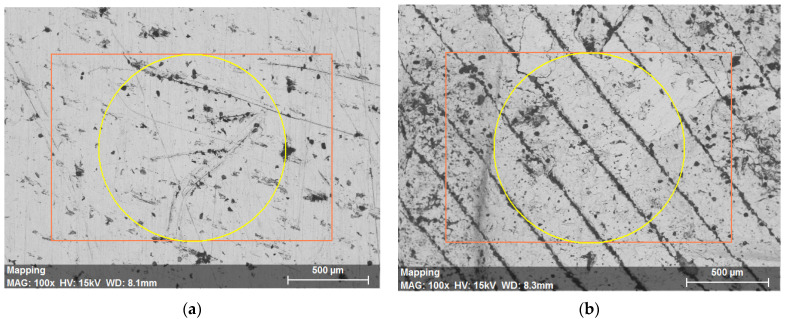
SEM images ×100: (**a**) Electrolytic part; (b) ADAM part. The colored graphs correspond to the areas bounded by the EDX software for the realization of the analyses.

**Figure 10 materials-15-05953-f010:**
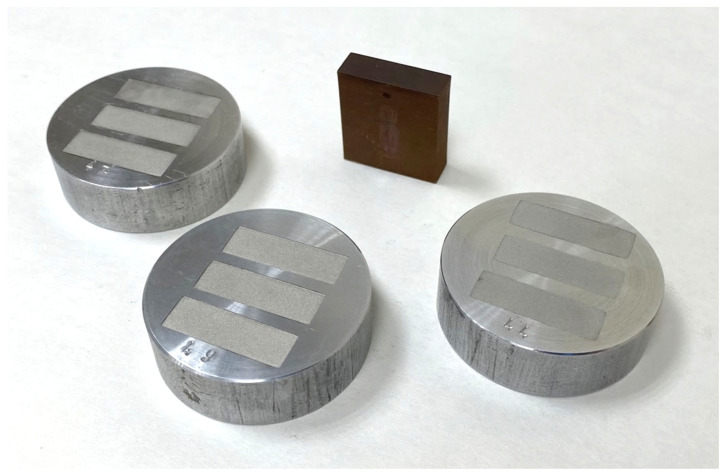
ADAM electrode and eroded aluminum workpieces.

**Figure 11 materials-15-05953-f011:**
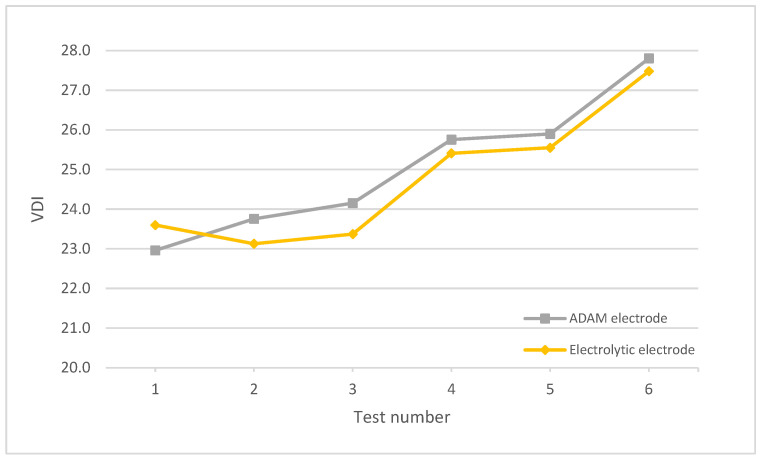
VDI of Workpieces obtained with electrolytic and ADAM copper electrodes.

**Figure 12 materials-15-05953-f012:**
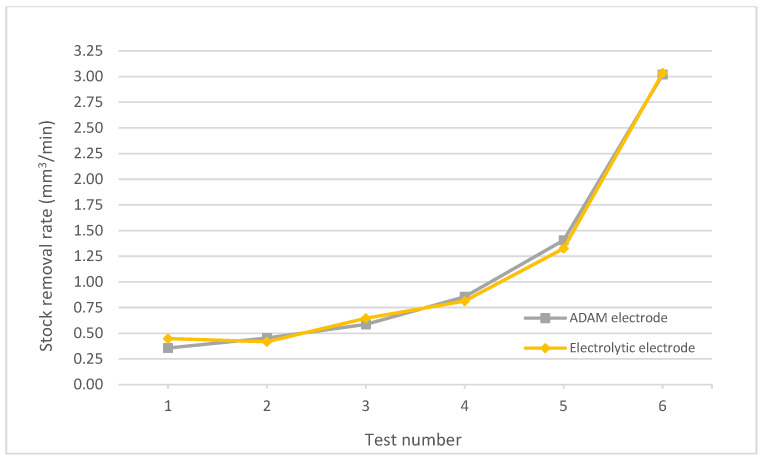
Stock removal rates for electrolytic and ADAM copper electrodes.

**Figure 13 materials-15-05953-f013:**
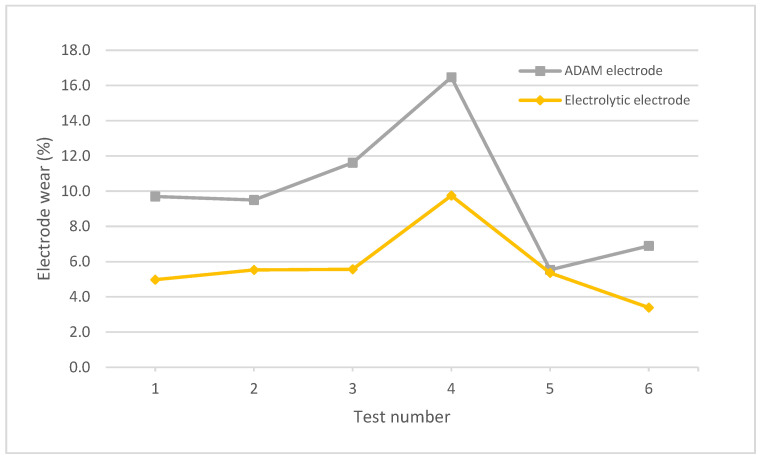
Wear rate of electrolytic and ADAM copper electrodes.

**Figure 14 materials-15-05953-f014:**
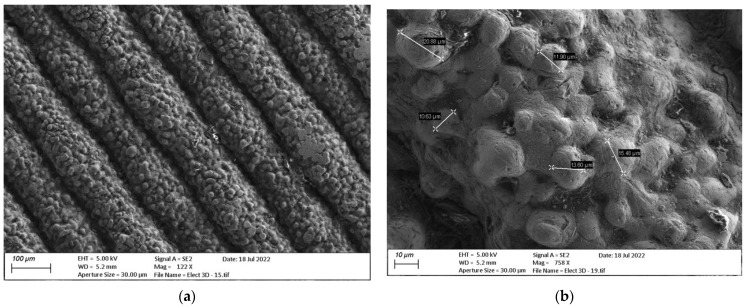
SEM images of ADAM copper electrode: (**a**) Lateral surface with layered and granulated structure (122× amplification factor); (**b**) Amplified (758×) lateral surface with particle size measurements.

**Figure 15 materials-15-05953-f015:**
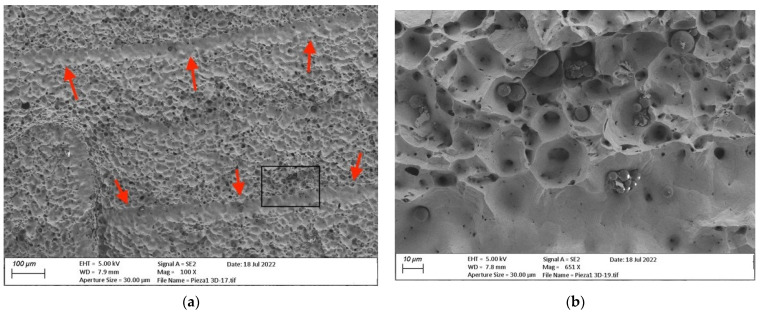
FESEM images of fractured ADAM copper cubic sample: (**a**) Fractured area (100× amplification factor) with voids and clearly marked regions between filaments (marked with arrows); (**b**) Detail (651×) where the inter-filament regions are clearly larger (larger voids) and not fractured, while the constriction effect during the fracture is clearly observed in the edges around the small holes (upper part of the image).

**Figure 16 materials-15-05953-f016:**
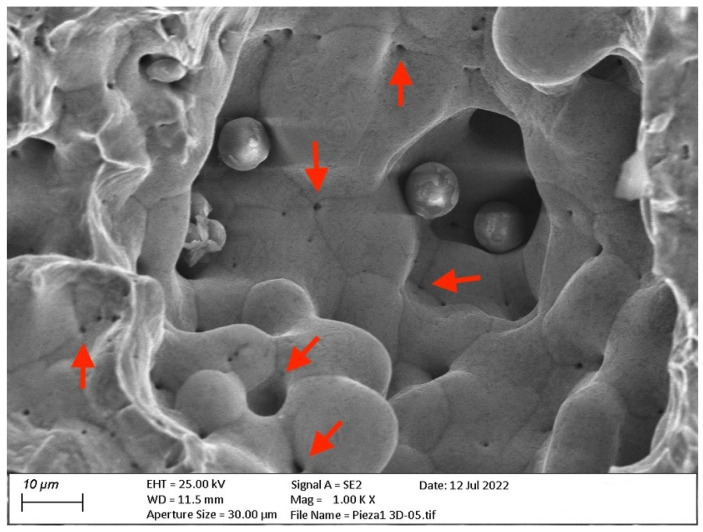
FESEM images of micro-voids (marked with arrows) and bonding interphases in fractured ADAM copper cubic sample.

**Table 1 materials-15-05953-t001:** Examples of potential procedures to produce EDM electrodes by AM.

Standard	Group	Some Examples
ISO/ASTM 52900	Direct AM	Powder-bed fusion (PBF) of metals Directed energy deposition Sheet metal lamination
	Indirect AM	AM + Electroforming, AM + cold spray, AM + thermal spray, AM + casting
ISO 17296-2	Single step by AM	Powder-bed fusion of metal Directed energy deposition Sheet metal lamination
	Multi-step by AM	Powder-bed fusion of composite plastic + copper infiltration Powder-bed fusion of composite plastic + debinding + sintering Binder jetting of composite + debinding + sintering Material extrusion of composite + debinding + sintering

**Table 2 materials-15-05953-t002:** Main printing parameters in Metal X printer.

Parameter	Value
Platform type	raft
Post-sintered layer height	0.129 mm
Roof and floor layers	8 layers (1.01 mm post-sintered)
Wall layers	4 layers (1.03 mm post-sintered)
Infill	triangular
Printed dimension	32.25 mm × 35.5 mm × 10.4 mm
Metal temperature	220 °C

**Table 3 materials-15-05953-t003:** Variable and fixed parameters of the EDM Cu–Al process.

Variable Parameters			Fixed Parameters	
Variable	Lower Limit	Upper Limit	Ionization Voltage (V)	−200
Average intensity	0.5 A (level 1)	2 A (level 3)	Erosion servo (V)	65
Impulse time (t_i_) (μs)	3.2	6.4	Return time (s)	0.3
Pause time (t_0_) (μs)	3.2	6.4	Work time (s)	0.3
Capacitor (C)	2	13	Flow level	15

**Table 4 materials-15-05953-t004:** Strategies defined for EDM Cu–Al.

	Strategy Number															
Parameter	1	2	3	4	5	6	7	8	9	10	11	12	13	14	15	16
Average intensity (level)	1	1	1	1	1	1	1	1	3	3	3	3	3	3	3	3
Impulse time (μs)	3.2	3.2	3.2	3.2	6.4	6.4	6.4	6.4	3.2	3.2	3.2	3.2	6.4	6.4	6.4	6.4
Pause time (μs)	3.2	3.2	6.4	6.4	3.2	3.2	6.4	6.4	3.2	3.2	6.4	6.4	3.2	3.2	6.4	6.4
Capacitor	2	13	2	13	2	13	2	13	2	13	2	13	2	13	2	13

**Table 5 materials-15-05953-t005:** Design variables and possible values for the optimization.

Design Variable	Possible Values in the EDM Machine	Possible Values for the Optimization
Average intensity (level)	From 1 to 16	1, 2 and 3
Impulse time (μs)	0.8, 1.6, 3.2, 6.4, 13, 25, 50, 100, 200, 400, 800, 1600, 3200	3.2 and 6.4
Pause time (μs)	0.8, 1.6, 3.2, 6.4, 13, 25, 50, 100, 200, 400, 800, 1600, 3200	3.2 and 6.4
Capacitor	From 1 to 39	From 1 to 13

**Table 6 materials-15-05953-t006:** Results of density measurements.

Parts	Density (g/cm^3^)
ADAM electrode	8.5198
Electrolytic electrode	8.9587
Aluminum workpiece	2.8207

**Table 7 materials-15-05953-t007:** Electrical conductivity of the electrodes.

Parts	σ_S_ (S/sq)	σ_V_ (S/m)
ADAM electrode	2.04 × 10^0^	2.58 × 10^2^
Electrolytic electrode	1.36 × 10^0^	1.94 × 10^2^

**Table 8 materials-15-05953-t008:** Quantification of elements on the surface of both electrodes.

	ADAM Electrode			Electrolytic Electrode		
Measurement	Cu (%)	C (%)	O (%)	Cu (%)	C (%)	O (%)
1	83.43	12.23	4.33	85.60	11.02	3.38
2	83.76	11.55	4.68	86.48	10.07	3.44
3	81.10	13.76	5.15	85.73	10.82	3.45
4	81.68	13.40	4.91	82.10	13.63	4.27
5	82.09	12.61	5.30	82.68	13.49	3.84
Mean value	82.41 ± 1.14	12.71 ± 0.89	4.87 ± 0.38	84.52 ± 1.98	11.81 ± 1.64	3.68 ± 0.38

**Table 9 materials-15-05953-t009:** Results of test EDM strategies Cu-Al.

	Design Variables (Process Parameters)				Response Variables		
Strategy	Average Intensity (Level)	Impulse Time (μs)	Pause Time (μs)	Capacitor (Level)	VDI	Electrode Wear Rate (%)	Stock Removal Rate (mm^3^/min)
1	1	3.2	3.2	2	25.48	7.681	0.486
2	1	3.2	3.2	13	26.29	10.514	0.592
3	1	3.2	6.4	2	23.60	4.979	0.448
4	1	3.2	6.4	13	24.92	10.765	0.447
5	1	6.4	3.2	2	22.62	7.290	0.479
6	1	6.4	3.2	13	25.01	10.705	0.591
7	1	6.4	6.4	2	23.63	5.294	0.392
8	1	6.4	6.4	13	24.58	10.432	0.391
9	3	3.2	3.2	2	23.37	5.568	0.646
10	3	3.2	3.2	13	25.41	9.753	0.813
11	3	3.2	6.4	2	23.25	5.781	0.442
12	3	3.2	6.4	13	25.27	9.934	0.568
13	3	6.4	3.2	2	23.33	6.274	0.626
14	3	6.4	3.2	13	25.16	9.731	0.809
15	3	6.4	6.4	2	23.13	5.531	0.417
16	3	6.4	6.4	13	25.49	10.150	0.551
17	3	6.4	3.2	6	23.65	15.475	0.046

**Table 10 materials-15-05953-t010:** Optimization algorithm results.

Preset VDI	Strategy	Design Variables (Process Parameters)				Corresponding Strategy	Response Variables		
		Average Intensity (Level)	Impulse Time (μs)	Pause Time (μs)	Capacitor (Level)		VDI	Electrode Wear Rate (%)	Stock Removal Rate (mm^3^/min)
23	Minimize electrode wear rate	3	6.4	6.4	2	15	23.13	5.487	0.415
	Maximize stock removal rate	3	3.2	3.2	2	9	23.37	5.524	0.642
24	Minimize electrode wear rate	1	3.2	6.4	2	3	23.60	4.940	0.446
	Maximize stock removal rate	3	3.2	3.2	3	18 (new)	23.59	6.285	0.630
25	Minimize electrode wear rate	2	3.2	3.2	4	19 (new)	24.51	7.511	0.565
	Maximize stock removal rate	3	3.2	3.2	13	10	25.41	9.676	0.809

**Table 11 materials-15-05953-t011:** Comparative results of new strategies 18 and 19.

	Preset VDI	Strategy	Design Variables (Process Parameters)				Result Type	Response Variables		
New Strategy			Average Intensity (Level)	Impulse Time (μs)	Pause Time (μs)	Capacitor (Level)		VDI	Electrode Wear Rate (%)	Stock Removal Rate (mm^3^/min)
18	24	Maximize stock removal rate	3	3.2	3.2	3	Estimated	23.59	6.285	0.630
							EDM test	25.55	5.013	1.326
19	25	Minimize electrode wear rate	2	3.2	3.2	4	Estimated	24.51	7.511	0.565
							EDM test	27.48	3.397	3.035

**Table 12 materials-15-05953-t012:** Results of ADM copper electrode EDM test.

		Design Variables (Process Parameters)				Response Variables		
Test Number	Strategy	Average Intensity (Level)	Impulse Time (μs)	Pause Time (μs)	Capacitor (Level)	VDI	Electrode Wear Rate (%)	Stock Removal Rate (mm^3^/min)
1	3	1	3.2	6.4	2	22.96	9.698	0.356
2	15	3	6.4	6.4	2	23.76	9.499	0.454
3	9	3	3.2	3.2	2	24.15	11.614	0.586
4	10	3	3.2	3.2	13	25.75	16.470	0.856
5	18	3	3.2	3.2	3	25.90	5.537	1.406
6	19	2	3.2	3.2	4	27.81	6.895	3.020

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
