# Peer review of "Evaluation of the Performance of Atomic Diffusion Additive Manufacturing Electrodes in Electrical Discharge Machining"

_materials, 2022, doi:10.3390/ma15175953_

Round 1

Reviewer 1 Report

1) The author needs to add more literature studies related to atomic diffusion additive manufacturing electrodes in electrical discharge machining.

2) The literature review is too general. Moreover, the literature gap is not discussed.

3) In a current form, a novelty of the conducted research is unclear.

4) In present article, it needs to add the main research contributions of the present research compared with the existing research.

5) In this article, it is necessary to add more experimental details related to density and conductivity measurements.

6) Similarly, it is necessary to add more experimental details related to surface roughness measurements.

7) The author wrote that “Regarding the ADAM electrode, the measured density was around 95.1% of the standard copper density (8.96 g/cm3), slightly lower than the Markforged specifications (98%).”

Therefore, it is necessary to provide the more reasonable and adequate explanation for the above experimental findings.

8) Why does ADAM electrode have high conductivity?

9) “In these cases, with a level 13 of capacitor and intensity level of 1, the wear increased on average by 68% compared to level 2 of capacitor. In the case of intensity level 3, this effect was also observed with an increase in wear of 71%.”

Similarly, it is necessary to provide the more reasonable and adequate explanation for the above research findings.

10) “The results were quite similar, with a slightly higher roughness for the ADAM electrode in all the tests except in test number 1, where the roughness was minimally lower by around 2.71%.”

Why does ADAM electrode have higher roughness?

Hence, it is necessary to provide the more reasonable and adequate explanation for the above research findings.

11) “In this case, the wear rate of the ADAM electrode was more pronounced than the one of the electrolytic electrode in practically all the tests except for number 5 (almost identical wear rates).”

Why was the wear rate of ADAM electrodes more pronounced than that of electrolysis electrodes?

So, it is necessary to provide the more reasonable and adequate explanation for the above research findings.

12) In Fig. 16, how to prove the existence of fracture characteristics?

13) In Fig. 7(b), the scale bar is missing.

14) The conclusion section needs to be improved.

15) The results are mainly presented by figures and tables. Nonetheless, some research results need to be reasonably and fully explained.

16) The limitations of the study are not considered.

Reviewer 2 Report

The paper deals with the Evaluation of the performance of Atomic Diffusion Additive 

Manufacturing electrodes in Electrical Discharge Machining.

The reviewer believes that the paper addresses a burning issue in manufacturing engineering, 

such as improving the quality of EDM machining processing.

Because this approach is innovative,

the reviewer suggests the paper for publication.

But, there are needs minor revision notes 

that are highlighted for the best presentation of the paper.

Comment 1

Line 6

The authors must add the Country.

Comment 2

The authors must format the paper according to the journal's instructions.

Delete the space in the page end (pages 1, 6, 11, 13, 14, 18 and 19).  

Comment 3

Extended text editing

Line 86

V. Kumar [3] 

The authors must replace

Meena et al. [3]

Line 89

A.K. Sahu et al 

The authors must replace

A. K. Sahu et al. 

Line 95

Uhlmann et al [6]

The authors must replace

Uhlmann et al. [6]

Line 100

T.Czelusmiak et al [7,8]

The authors must replace (replace m to n)

T. Czelusniak et al. [7,8]

Line 103

E. Uhlmann et al [9]

The authors must replace 

E. Uhlmann et al. [9]

Line 108

Uchida et al [10]

The authors must replace 

Uchida et al. [10]

Lines 109 - 110

D. Yanagida et 

al [11]

The authors must replace 

D. Yanagida et 

al. [11]

Line 121

Alamro et al [13] 

The authors must replace 

Alamro et al. [13] 

Line 130

S.K. Padhi et al [14].

The authors must replace

S. K. Padhi et al. [14].

Line 138

C. Sánchez et al [15].

The authors must replace

C. Sánchez et al. [15].

Line 144

by Q.X. Wu et al [17] 

The authors must replace

by Q. X. Wu et al. [17] 

Line 381, Table 6, Table 10, Table 11, Table 12, Figure 13

was converted to mm3

The authors should format the "3".

Comment 4

Table 2

8 ud

The authors must define (give more details) the "ud".

Comment 5

Line 218

that correspond to 2.7%.

The authors should check if the value 2.7% is right (57.33 to 55.68 is not 2.7%).

Comment 6

Line 310

Average Intensity: 0.5 A (level 1) – 2 A (level 3) 

The authors must define (give more details) the level 2.

Comment 7

Lines 369 - 370

always keeping the VDI value below 24.

Lines 293 - 295

VDI grades from 16 to 22 (copper 

electrode on steel workpiece) were established as reference for its typical applications in 

mold manufacturing, where EDM process is very useful.

The authors must explain why they did not use the value 22 in the Lines 369 - 370 (instead the 24).

Comment 8

Table 6, Table 9, Table 11, Figure 17 

The Table or Figure must be accompanied on the same page as the Table or Figure's title.

Comment 9

Figure 10

The authors must explain why there is a different machining area (smaller - middle area) in the specimen 2.

Comment 10

Table 10

Major problem

Average 

intensity 

(level)

The authors must define the level 2.

Comment 11

Lines 621 - 622

The results of density and electrical conductivity of the electrode manufactured with 

the ADAM technology showed similar values to electrolytic copper,

The authors must explain the meaning of "similar" (33.33%?).

Density

ADAM electrode 8.5198 Electrolytic electrode -------Result: 8.9587 = 4.89%

Electrical conductivity of the electrodes 

σS ADAM electrode 2.04 Electrolytic electrode 1.36 --------Result: 33.33%

σv ADAM electrode 2.58 Electrolytic electrode 1.94 --------Result: 24.8% 

Reviewer 3 Report

The authors present the development of a copper electrode manufacturing approach with Atomic Diffusion Additive Manufacturing (ADAM) for Electrical Discharge Machining (EDM) and its performance compared with a conventional electrolytic copper electrode.  Authors design of EDM experiments and conducted optimizations using genetic algorithms were carried out to establish a comparative framework for both electrodes and evaluate the electrode wear rate, the roughness of the workpiece and the rate material removal with both electrodes.

The article is very interesting and good fits the profile of the journal. I have no fundamental objections. I think it will be suitable for publication after corrections.

Strengths

In my opinion, the strong point of this article is the innovative subject matter and the use of modern research equipment for tests

Noticed errors

1.    The authors use a mix of UK English and US English. It would be appropriate to choose one language and stick to it.

2.    Figure 1: The drawing on the right is almost illegible. You should change the contrast/brightness.

3.    The state of the issue presented in the paper lacks a broader analysis and presentation of the research gap resulting from this analysis. This should be completed.

4.    Table 2. What is ud ?

5.    Figure 21: The drawings are almost illegible. You should change the contrast/brightness.

6.    Section 2.6 Roughness measurement needs to add information on signal filtering and the filters used in the roughness measurements.

7.    Figure 10 adds nothing to the work and is unnecessary.

Small errors

1.       Line 257. The variable R and l should be written in italics

2.       Line 302 and 303. Between value and unit must be space. Applies to the entire work.

3.       Line 381. Is: mm3; should be: mm3

4.       Table 9, 10, 11 and 12. Is: (mm3/min); should be: (mm3/min)

5.       Wrong pagination of pages 22/23,